# The Dengue ED3 Dot Assay, a Novel Serological Test for the Detection of Denguevirus Type-Specific Antibodies and Its Application in a Retrospective Seroprevalence Study

**DOI:** 10.3390/v11040304

**Published:** 2019-03-27

**Authors:** Heidi Auerswald, Leonard Klepsch, Sebastian Schreiber, Janne Hülsemann, Kati Franzke, Simone Kann, Bunthin Y, Veasna Duong, Philippe Buchy, Michael Schreiber

**Affiliations:** 1Department of Virology, Bernhard Nocht Institute for Tropical Medicine, Bernhard Nocht Str. 74, 20359 Hamburg, Germany; hauerswald@pasteur-kh.org (H.A.); leonardklepsch@gmail.com (L.K.); sebastian@schryber.com (S.S.); j.huelsemann91@web.de (J.H.); Kati.Franzke@fli.de (K.F.); simone.kann@medmissio.de (S.K.); 2Virology Unit, Institut Pasteur in Cambodia, 5 Monivong Boulevard, 12201 Phnom Penh, Cambodia; ybunthin94@yahoo.com (B.Y.); dveasna@pasteur-kh.org (V.D.); philippe.x.buchy@gsk.com (P.B.); 3GlaxoSmithKline, Vaccines R&D, 23 Rochester Park, Singapore 139234, Singapore

**Keywords:** dengue virus, serology, serotype, antigen assay, domain 3, E protein, seroprevalence

## Abstract

There are four distinct antigenic serotypes of dengue viruses (DENV-1-4). Sequential infections with different serotypes lead to cross-reactive but also serotype-specific neutralizing antibody responses. Neutralization assays are considered as gold standard for serotype-specific antibody detection. However, for retrospective seroprevalence studies, access to large serum quantities is limited making neutralization assays well-nigh impossible. Therefore, a serological test, wasting only 10 µL serum, was developed using fusion proteins of maltose binding protein and E protein domain 3 (MBP-ED3) as antigens. Twelve MBP-ED3 antigens for DENV-1-4, three MBP-ED3 antigens for WNV, JEV, and TBEV, and MBP were dotted onto a single nitrocellulose strip. ED3 dot assay results were compared to virus neutralization and ED3 ELISA test results, showing a >90% accordance for DENV-1 and a 100% accordance for DENV-2, making the test specifically useful for DENV-1/-2 serotype-specific antibody detection. Since 2010, DENV-1 has replaced DENV-2 as the dominant serotype in Cambodia. In a retrospective cohort analysis, sera collected during the DENV-1/-2 endemic period showed a shift to DENV-2-specific antibody responses in 2012 paralleled by the decline of DENV-2 infections. Altogether, the ED3 dot assay is a serum-, time- and money-saving diagnostic tool for serotype-specific antibody detection, especially when serum samples are limited.

## 1. Introduction

The mosquito-borne dengue virus (DENV) causes infection with various clinical outcomes ranging from asymptomatic infections over self-limiting febrile illness, to severe complications like hemorrhage and/or circulatory shock. A particular characteristic of DENV is the classification into four serotypes designated as DENV-1, DENV-2, DENV-3 and DENV-4. Natural infections occur by the bite of a DENV-infected mosquito, leading to cell-mediated response as well as to a broad humoral immune response. Although homotypic reinfections have been recently documented [1] a primary infection with one of the four serotypes is usually considered to render a lifelong protective homologous immune response, preventing a symptomatic infection with the same serotype [2,3]. However, this immunity does not provide long-time protection against infection with another serotype as the cross reactive antibodies produced from the first infection are short-lived and have poor or non-neutralizing activity against other serotypes [4,5]. These non-protective antibodies are hypothesized to increase the number of infected cells, leading to an augmented infection and consequently a possibly more severe clinical outcome, a phenomenon known as antibody-dependent enhancement [6,7]. Cross-reactivity does not only occur between the DENV serotypes but also with other flaviviruses [8,9]. This is a major concern especially in regions where different DENV serotypes co-exist and/or other flaviviruses co-circulate, and multiple infections are common [10]. It also restrains the development of effective DENV vaccines that must induce a protective and balanced immune response against all four serotypes.

The direct virus diagnostic in the clinical specimen is performed by the detection of the virus antigen (NS1) or viral nucleic acids (RT-PCR), but this is only applicable during the acute phase of infection, as the virus is rapidly cleared by a patient’s immune response [11]. Various serological assays are able to detect DENV antibodies including enzyme-linked immunosorbent assays (ELISAs), hemagglutinin inhibition assay (HIA) and neutralization assays (NTs) like the plaque reduction neutralization test (PRNT). The common NT principle is based on the fact that neutralizing antibodies block the virus entry or the fusion with the host cell membrane [12,13]. Although NTs are the most serotype-specific serological assays, they are not widely used due to several practical limitations. The main disadvantage of NTs is the necessity of using specific biosafety containment measures in non-endemic countries in order to grow the infectious virus. Additionally, these assays are laborious, time-consuming and most importantly, require a significant volume of serum as the test has to be performed with several viruses. Neutralization assays are poorly adapted for high throughput studies, making the NT-application in large-scale surveillance and vaccine control studies extremely fastidious [14,15]. Therefore, there is a need for a quick and simple test to study DENV serotype-specific antibody responses in patient sera.

As for other flaviviruses, the DENV particle is formed by the structural proteins capsid (C), precursor membrane (prM) and the envelope glycoprotein (E). The surface of immature virions is covered by prM-E heterodimers and mature virions are coated with M-E heterodimers. Both virion structures and therefore both proteins are targets of a broad range of antibodies. These proteins are integrated into the virus membrane to facilitate cell attachment, receptor binding as well as fusion of virus and cell membranes [16]. The majority of antibodies are partially or fully cross-reactive between DENV E and only a minority of the antibody response in naturally infected patients was shown to be directed against protein E with a high DENV serotype-specificity [17]. The E protein is mainly divided into three domains, designated ED1, ED2 and ED3, which are followed by the stem and the membrane anchor region. DENV neutralizing antibodies derived from patients’ sera are found to be directed against all three E domains, but ED3 is the target for the most potent serotype-specific neutralizing antibodies [18,19]. Besides the epitopes for serotype-specific antibodies, ED3 additionally contains epitopes for serocomplex-reactive antibodies that recognize viruses from more than one DENV serotype [20,21,22]. As ED3 incorporates so many epitopes for a broad range of antibodies, this domain is frequently used as a universal DENV antigen in diverse serological assays [23,24,25,26]. However, studies on the natural immune response to DENV based on the characterization of human monoclonal antibodies (mAb) against the E protein demonstrated that the majority of the antibodies was fully or partially cross-reactive and only 1% of the anti-E human mAbs was found to be serotype-specific [6]. Additionally, human mAb were predominantely directed to E proteins on the virion surface and not to recombinant E [27]. Thus, recombinant E or ED3 antigens are rarely used for the detection of serotype-specific anti-DENV-antibodies from natural infection [28,29]. Altogether, the specificity of ED3-derived antigens for the determination of DENV serotypes and their application in a serological test seems to be controversial and needs further clarification.

In this study, we present results of a simple ED3 dot assay using three different recombinant antigens, called ED3, ED3s and mED3. Results were directly compared with the serotype-specific antibody responses determined with a neutralization test, the gold standard for serological flavivirus diagnostic, for a subset of 85 sera. Additionally, the ED3 antigens were evaluated for their serotype-specificity in an ELISA by using a subset of 67 sera from dengue patients. Altogether, the ED3 dot assay was used for the detection of DENV serotype-specific antibody responses in 1099 sera from Vietnam, Colombia and Cambodia. In order to assess the practicality of the ED3 dot assay in the context of a seroprevalence study, we used a subset of 697 sera obtained from Cambodian patients collected in 2010 and 2012 to study DENV-1 and DENV-2 serotype-specificities during the DENV-1/-2 endemic period.

## 2. Materials and Methods 

### 2.1. Cell Culture

VeroB4 cells (DSMZ no.: ACC 33) were used for DENV cultivation as well as for titration of virus culture supernatants and the foci reduction neutralization test (FRNT). VeroB4 cells were cultured in Dulbecco’s modified Eagle’s medium (DMEM; PanBiotech, Aidenbach, Germany), supplemented with 10% FCS (PanBiotech, Aidenbach, Germany), 2 mM glutamine (PanBiotech), 1% antibiotic solution (5000 U/mL penicillin, 5.000 µg/mL streptomycin; Gibco, USA) at 37 °C and 5% CO_2_.

### 2.2. Viruses

The following DENV strains were used in all neutralization tests: DENV-1 Hawaii (Genbank: KM904119), DENV-2 16621 (Genbank: U87411), DENV-3 H87 (Genbank: M93130) and DENV-4 H241 (Genbank AY947539). All viruses were cultivated in VeroB4 cells. Virus culture supernatants were harvested by centrifugation and concentrated with 8% PEG 8000 overnight at 4 °C. Viruses were precipitated after that by centrifugation at 1500× *g* for 30 min, the virus containing pellets were suspended in DMEM and stored at −80 °C until use. For cloning of the ED3 antigens we used the following viruses: DENV-1 West Pac strain (GenBank: U88535), DENV-2 TH/BID V3357/1964 (GenBank: GQ868591), DENV-3 H87 (GenBank: M93130), DENV-4 PH/BID V3361/1956 (GenBank: GQ868594), WNV Uganda (Genbank: M12294), JEV Nakayama (GenBank: EF571853) and TBEV Neudörfl (GenBank: U27495).

### 2.3. Serum Specimens

In total, 1399 sera were analyzed in this study, including 1099 DENV-positive (2nd serum samples) and 300 DENV-negative control samples. Serum samples from Cambodia (*n* = 1014) were provided by the Institut Pasteur in Cambodia (IPC). Other DENV-positive sera were from Colombia (*n* = 28) and Vietnam (*n* = 57, provided by D. Ludolfs [26]). The control sera were divided into DENV-negative/Japanese Encephalitis Virus (JEV)-positive samples (*n* = 97), and DENV-negative and JEV-negative sera (*n* = 203). Sera were formerly characterized by RT-PCR, IgM MAC-ELISA and HIA [17]. The use of stored and partially anonymized samples for research purposes and in particular for the development of new diagnostic tools was approved by the Cambodian National Ethics Committee. Samples from Vietnam were obtained from healthy Vietnamese people during studies on amoebiasis and dengue fever at the city of Hue in 1999 [30]. The collection and use of Vietnam serum samples of patients with dengue fever was approved by the Ethics Committee of the Ärztekammer Hamburg (WF-024/11). The presence of dengue antibodies was previously confirmed by indirect immunofluorescence [28]. Serum samples from Colombia were collected from patients who were tested positive for dengue by RT-PCR in a study at the Hospital Rosario Pumarejo de Lopez in Valledupar, Colombia, that was approved by the local ethic commission.

### 2.4. Maltose Binding Protein-ED3 Fusion Proteins

Viral RNA was extracted using QIAamp Viral RNA Mini Kit (Qiagen, Hilden, Germany). The RNA was reverse-transcribed using Revert Aid H Minus M-MuLV Reverse Transcriptase (ThermoFisher, Darmstadt, Germany) according to the manufacturer’s instructions. The ED3 coding region was amplified from cDNA with specific primers and Phusion High-Fidelity DNA Polymerase (New England Biolabs, Frankfurt am Main, Germany). The forward primers were designed to introduce a BamHI restriction site at the 5′-end and the reverse primers contained a HindIII restriction site at the 3′;-end of the PCR product. After BamHI and HindIII digestion, the PCR fragment was cloned into pMAL-p4x vector (New England Biolabs) downstream to the malE gene [31]. Since this gene encodes the malEss signal sequence that directs the maltose binding protein (MBP) to the bacterial periplasm, the reducing environment of the cytoplasm is bypassed to form stable disulfide bonds between the conserved cysteine amino acids present in ED3. All MBP constructs were sequenced to verify the respective identity (LGC genomics, Berlin, Germany). For expression of the MBP-ED3-fusion proteins, pMAL-p4x plasmids containing the insert of choice were transformed into E. coli BL21(DE3) (Promega, Mannheim, Germany) cells. A volume of 2.7 L of dyt-medium supplemented with 300 µg/mL ampicillin, 0.5 mM IPTG and 0.2% glucose was inoculated 1:100 with an overnight culture and incubated at 37 °C in a horizontal shaker (GFL3031, GFL, Burgwedel, Germany) at 180 rpm. The bacteria were harvested after four hours via centrifugation at 10,000 rpm for 3 min (Beckman Coulter Avanti J-26 XP, rotor JA-14) and were suspended in lysis buffer (20 mM Tris pH 7.4, 200 mM NaCl, 1 mM EDTA, 5 mg/mL lysozyme). Bacterial lysis was carried out by sonication on ice and clear lysates were obtained after centrifugation for 30 min at 4 °C, 15,000× *g* (Eppendorf 5810R, rotor FA-45-6-30). The MBP-ED3-fusion proteins were purified from clear lysates using amylose resin affinity chromatography according to the manufacturer’s recommendations (New England Biolabs). The protein was eluted with maltose containing buffer (20 mM Tris/HCl pH 7.4; 200 mM NaCl; 1 mM EDTA; 10 mM maltose) and the resulting fractions were analyzed on a 10% SDS-PAGE. Pooled and concentrated fractions containing the antigen of interest were purified via size exclusion chromatography. Therefore, a FPLC System (Pharmacia, Uppsala, Sweden) with a prepacked HiLoad 16/60 Superdex 75 column (GE Healthcare, Solingen, Germany) was used. Recombinant antigens were eluted from the column using an aqueous buffer solution containing 100 mM Tris pH 8, 300 mM NaCl, 1% acetonitrile. Eluted proteins were photometrically detected at 280 nm and fractions were further analyzed on a 10% SDS-PAGE. Matching fractions were pooled and concentrated to a final concentration of 1 mg/mL. The mED3 antigens were prepared by adding SDS to a final concentration of 1% to the respective ED3 constructs, heated to 95 °C for 10 min and dotted, together with the other antigens, onto nitrocellulose strips. After the dotting procedure the test strips were air dried, sealed into plastic bags and stored at 8 °C until use.

### 2.5. Foci Reduction Neutralization Test (FRNT)

Flat-bottom 96-well plates were seeded with VeroB4 cells (4 × 104 per well) 24 h before infection. Patients’ sera were heat inactivated at 56 °C for 30 min. Two-fold serial serum dilutions starting at 1:10 were prepared in DMEM and added to equal volumes of virus representing 50 foci forming units (ffu) per well. Virus-serum mixtures were incubated for one hour at 37 °C. After incubation the virus-serum mixtures were added to VeroB4 monolayers and incubated for one additional hour at 37 °C and 5% CO_2_. After infection, the virus-serum mixtures were removed and a semi-solid overlay (0.8 % methyl cellulose, DMEM, 10% FCS) was added. Microplates were incubated three days at 37 °C. After that, formaldehyde solution (3.7% in PBS (137 mM NaCl, 2.7 mM KCl, 10 mM Na_2_HPO_4_, 1.8 mM KH_2_PO_4_, pH 7.4)) was added and incubated for 20–30 min at room temperature. The overlay-formaldehyde mixture was decanted and the plates were washed with PBS followed by a PBS/0.5% Triton X-100 treatment for 20 min. After additional washing the wells were coated with 10% FCS in PBS for 30 min. Virus infected cells were stained with anti-DENV mouse hyperimmune ascites fluids (IPC, Phnom Penh, Cambodia) for one hour. After washing, the plates were incubated with anti-mouse IgG antibody conjugated to horseradish peroxidase (Bio-Rad, Hercules; CA USA), for one hour, washed again and then incubated 20 min with TMB substrate (Mikrogen Diagnostic, Neuried, Germany). The foci were counted immediately and the endpoint titers were expressed as reciprocal of the highest serum dilution showing ≥ 90% reduction in foci counts (FRNT90 titer) compared to wells without serum. All sera were tested in triplicate. The DENV serotype was considered as the virus that induced the highest FRNT90 titer. A picture of an original test plate is shown in the Appendix A as Figure A1.

### 2.6. ED3 Enzyme-Linked Immunosorbent Assay (ELISA)

The antigen concentration was adjusted to 2 µg/mL with bicarbonate/carbonate buffer (100 mM, pH 9.2). Afterwards, 100 µL of the diluted antigen were applied to each well of a 96-well plate (Maxisorb, Greiner Bio-One, Frickenhausen, Germany), sealed and incubated at 8 °C overnight to coat the plates with the respective antigen. For testing, sera were diluted 1:100 in PBS containing 5% low fat milk, and purified MBP protein was added to a final concentration of 75 µg/mL. The serum-MBP mixture was incubated at 8 °C overnight. Before adding the serum-MBP mixture, each well of the coated plate was filled with 300 µL blocking buffer (PBS, 5% low fat milk) to cover unspecific binding sites. After 1 h at room temperature, the plates were washed three times with PBST (PBS, 0,05% Tween 20). Into each well 100 µL of the serum-MBP mixture was added and incubated at room temperature for 1 h. After that, the mixtures were discarded and plates were washed three times with PBST. Following, 100 µL of HRP-conjugated goat anti-human anti-IgG antibody (Bio-Rad, München, Germany) diluted 1:1000 were added to each well and incubated for 1 h. Plates were washed three times with PBST, followed by three wash cycles with PBS. To each well 50 µL TMB substrate (KPL SureBlue, medac, Wedel, Germany,) was added. The color reaction was stopped after 10 min by adding 50 µL 1N H_2_SO_4_ per well. The intensity of the color reaction was measured at 450 nm.

### 2.7. ED3 Dot Assay

To deplete antibodies directed against MBP, sera were diluted 1:100 in blocking buffer (5% low fat milk in PBST), containing 50 µg/mL MBP and were incubated at 8 °C overnight. Purified antigens were transferred into a 394-well microplate and dotted on 2.7 mm wide and 115 mm long nitrocellulose strips (Schleicher & Schuell BA85) by using a custom made 24-pin dot blotter with stainless steel pins (20 mm long, 1.2 mm diameter) with a 2.5 mm stainless steel ball ending. Using this dot blotter 0.5 µL of the antigen solution (1 mg/mL) was transferred to the nitrocellulose test strip for each antigen dot. The test strips were transferred to a 30-well incubation tray (Viramed, Planegg, Germany) and were incubated with blocking buffer for 1 h at room temperature. The blocking buffer was discarded and 1 mL of the pre-incubated serum solution was applied to each strip. The stripes soaked with serum solution were incubated on a shaker for 2 h followed by three wash cycles with PBST. The detection of bound human antibodies was carried out with an anti-human IgG antibody conjugated to horseradish peroxidase (Bio-Rad, Hercules; CA, USA) diluted 1:1000 in blocking buffer. After 1 h of incubation the test strips were washed again three times with PBST buffer followed by three wash cycles with PBS. Bound antibodies were visible as dots after 10–20 min incubation with 4-chloro-1-naphtol (4 CN) solution, a freshly prepared mixture of 200 mL PBS, 100 µL H_2_O_2_ (30% stock solution, Merck, Darmstadt, Germany) and 40 mL 4CN solution (0.3% in methanol). The intensity of the dots was analyzed by an in-house made purpose-built dot analyzing software (BlotLog). We have developed the BlotLog software especially for our test strips to allow a more convenient scanning procedure in comparison to the universal UN-SCAN-IT graph digitizer software (Silk Scientific, Orem, UT, USA), which also allows such kind of dot analysis. Using BlotLog, for each dot, a sample size representing 100 pixels centered within the dot was averaged to calculate the gross dot intensity. The sampling size was about 50% of the dot’s total pixel count. Since the background intensity of a test strip could vary slightly from top to bottom, the background intensity of the test strip around each dot was linearly interpolated to calculate the background intensity for each individual dot. The gross dot intensity subtracted by the individual background resulted in the net dot intensity (I_n_). The BlotLog derived I_n_ values were then grouped respective to their antigens (ED3, ED3s, mED3 and controls) and analyzed by RStudio to decide whether a serum sample showed single, double, triple or quadruple responses to one of the ED3, ED3s and mED3 antigen sets. The RStudio calculation was repeated and factors fine-tuned until the automated results matched the results when manually examining a set of representative test strips. For denatured mED3, I_n_(max) = 193 was set to 100% and the cut-off was set to 4.8%. For native ED3 and ED3s, I_n_(max) = 234 and 222 was set to 100%, respectively, with a cut-off set to 25%. Finally, all dot results were exported in Excel format for further analysis of the data.

## 3. Results

### 3.1. Recombinant ED3 Antigens

Several C- and N-terminally truncated constructs of DENV-1 ED3 were tested as maltose-binding protein (MBP) fusion proteins for their capacity to bind DENV antibodies (Figure 1A). In contrast to all N-terminally truncated antigens, the full ED3 domain (construct 302–399) reacted positively to DENV-infected patients’ sera. Additionally, ED3 constructs including amino acid 302 and the complete stem region (construct 302–443) or parts of the stem (constructs 302–435, 302–429, 302–418, and 302–410) reacted positively. Out of these four ED3-stem constructs, the 302–418 construct showed the strongest reactivity to DENV-infected patients’ sera. For serological testing we therefore selected the two antigens 302–399 and 302–418, designated ED3 and ED3s, respectively.

For the ED3 dot assay the ED3-MBP fusion proteins for different types of viruses (Figure 1B) were all expressed in *Escherichia coli* bacteria (strain DH5α) and purified by affinity and size-exclusion chromatography. The purity of the proteins used for the antigen assay was examined via SDS-PAGE followed by sensitive silver staining (Figure 1C). Additionally, purified ED3 antigens were modified by denaturation, designated mED3. In previous test runs, the denatured version of ED3s (mED3s) did not show any serotype-specific difference compared to mED3. Therefore, the mED3s antigen version was not included into the design of the ED3 antigen array (Figure 1D). In addition to DENV-1-4 ED3, ED3s and mED3 antigens, the ED3 domains of three other flaviviruses, West Nile virus (WNV), Japanese encephalitis virus (JEV) and Tick-born encephalitis virus (TBEV) were cloned and expressed for control purposes. Furthermore, the MBP antigen itself was produced and purified the same way as the other antigens and served as an additional control on the test strips.

### 3.2. Sera Used in the Study

For the study of anti-DENV serotype-specific antibody responses we have used 1099 DENV-positive 2nd serum samples and 300 DENV negative controls (Figure 2). All these sera were tested with the ED3/ED3s/mED3 dot assay by using 10 µL of serum, since only limited volumes of sera were available. From a subset of 148 sera, volumes were available to perform FRNT neutralization and ED3 ELISA tests.

### 3.3. ED3 Dot Assay Responses and Comparison to Serotype-Specifc Results Obtained by FRNT and ED3 ELISA

Next we have compared results from the ED3 dot assay with two tests, ED3 ELISA and FRNT. In Figure 3, results are exemplarily shown for each DENV. Representatives of each serotype were chosen after being tested via FRNT. Serotype-specificity was determined by the highest serum dilution that neutralized the respective DENV by >90%. Results of the serotype-specificity obtained by FRNT were reproduced by the ED3 dot assay (Figure 3, black bars). In Figure 3A, mED3 response to DENV-3 was below cut-off level (4.7%) with a DENV-1 response > 50%. In Figure 3C,D, responses to DENV-3 and DEMV-4 was 12.1% and 16.3%, respectively, showing values clearly above cut-off level. Reactivity to all other mED3 antigens was below 0.7%. Thus, all four sera were classified as mED3 single positives.

The ED3 ELISA also confirmed the ED3 dot assay and FRNT results for each of the four sera by showing the highest optical densities for the DENV serotype-specific antibody response, formerly detected with the other two assays.

A more detailed comparison of ED3 dot assay and FRNT, using DENV-1-4 FRNT-reference viruses, was carried out based on results obtained with 85 sera tested in both assays (Table 1). A second subset of 67 sera was tested with the ED3 dot assay and the results were compared to those of the ED3 ELISA (Table 2). Overall, the results of the ED3 dot assay and ED3 ELISA matched for 92.5% of the analyzed sera, and the agreement between ED3 dot assay and FRNT was 88%. The best matching results were obtained for DENV-1 (93.5% FRNT; 92% ELISA) and DENV-2 (100%). Thus, the ED3 dot assay using the chosen reference strains seems to be well suited for seroprevalence studies in regions where DENV-1 and DENV-2 are highly prevalent.

### 3.4. Reactivity of ED3 Antigens on Test Strips

On the original test strips we have dotted mED3, ED3s, ED3, controls and additionally eight other non-ED3 related antigens. For a more transparent presentation of the data the non-ED3 related dots were removed from the original figure. The original figure is shown in the Appendix A (Figure A2).

Reactivity of DENV-positive sera against ED3 and ED3s antigens showed under the experimental conditions no serotype-specific reactivity (Figure 4, serum dilution 1:100). Most of the ED3s or ED3 antigens reacted to sera and especially all the ED3 antigens were highly reactive. ED3 for JEV and WNV also showed strong responses whereas sera were negative to TBEV. Two sera (no. 10 and 12) showed a weak response to the MBP control but clearly distinguished between the four mED3 antigens. In contrast to ED3 and ED3s, the mED3 antigen panel reacted more specific and we observed single (serotype specific), double, triple (partially cross-reactive) positive dots and also clearly negative dot results. Also quadruple (fully cross-reactive) responses were detected by mED3 (Figure 4, no. 15). These type of sera reacted also strongly to the other eight DENV-1-4 ED3 and ED3s antigens. Thus, in these overall reactive sera no serotype-specific response could be detected by mED3. In Table 3 direct and indirect diagnostic parameters of the sera used in Figure 4 are shown.

Test strips were used to analyze antibody responses in 1099 DENV-positive and 300 DENV-negative control sera. The dots, numbers of positively reacting antigens per serum were counted for each set of the DENV-1-4 antigens. As shown in Figure 5A, most of the sera (*n* = 857) reacted to all four ED3 antigens (quadruple-positives). Single- or double-positive responses to the ED3 antigen set were low, but serum numbers for single- and double-positives increased in the ED3s and mED3 antigen sets. All 300 DENV-negative sera (Figure 5A, controls), including 97 sera from JEV confirmed patients, showed no reactivity to any of the four mED3 antigens. Single dot positivity was more frequently observed in the group of mED3 antigens (58%, 642/1099 sera) compared to ED3s antigens (11%, 122/1099) and ED3 antigens (5%, 59/1099). The number of double-positive dot responses was also higher in the mED3 group (195/1099) compared to ED3s (181/1099) and ED3 (50/1099). Thus, the mED3 antigen set detected more single- and double-positive sera compared to ED3 and ED3s. When considering all the single- and double-positive dot responses obtained by the ED3 dot assay, we observed an overall trend towards more serotype-specific antibody reactions using mED3. However, testing with mED3 led to a 16% (178/1099 sera) rate of completely negative reactions (mED3-negative). Thus, denaturation yielded in lower test sensitivity but increased serotype-specificity. Analysis of these 178 mED3-negative serum samples by ED3s led to 29/178 (16%) ED3s single- and 48/178 (27.5%) double-positive results (Figure 5B). Nevertheless, 23/178 (13%) of these sera remained mED3/ED3s non-reactive. Analysis of these 23 mED3/ED3s-negative serum samples by ED3 led to 8/23 (35%) single- and 2/23 (9%) double-positive results.

By combining all single- and double-positive results, we identified in total 924 (84%) sera that reacted against a single or against two different DENV-1-4 antigens (Figure 5C). The results obtained with ED3s antigens improved the serotype-specific rate by 7% (Figure 5B, 29 + 48/1099 sera) and the use of ED3 only increased the number of serotype-specific results by 1% (Figure 5B, 8 + 2/1099 sera). From these results we concluded that the best antigen combination would be an antigen array containing two sets of DENV-1-4 antigens represented by ED3s and mED3. 

### 3.5. Retrospective Analysis of the DENV Serotype-Specific Antibody Response in CAMBODIA in 2010 and 2012

In the group of DENV-positive sera (*n* = 1099) tested by the ED3 dot assay two large subsets were present, one from Cambodian patients in 2010 and the other from Cambodian patients in 2012. From these two subsets 296 sera from 2010 and 401 sera from 2012 showed single- and double-positive antibody responses to mED3. In Figure 6A,B the monthly distribution of positive serum numbers is given for 2010 and 2012 (significant changes from 2010 to 2012 are marked in red).

The assay revealed that DENV-1 serotype-specific antibodies were predominant in the samples from 2010 whereas antibody specificities against the other three DENV serotypes were about 50 % less frequent (Figure 6C, white bars). In 2010 most of the sera showed a mED3 response against DENV-1 which slightly decreased in 2012 (Figure 6C, black bars) same as the DENV-1 + 3 responses (*p* = 0.047). On the other hand, antibody responses against DENV-2 increased significantly in 2012 (*p* = 0.003) in parallel with responses to DENV-2+4 (*p* = 0.003) (Figure 6C). Overall, we observed decreasing seroprevalence changes for DENV-1 and DENV-3 from 30 to 25% and from 16 to 11%, respectively. In contrast, DENV-2 specific responses significantly increased from 16 to 25%, and for DENV-2+4 from 3 to 8.5%. The frequencies for DENV-4 serotype specific antibodies were nearly constant in both years (14 and 15 %, respectively).

Data from the Cambodian national surveillance program for DENV [32] showed that over a period of four years, DENV-1 had replaced DENV-2 with a clear dominance of DENV-1 infections in 2012 (Figure 7). In 2010, a drastic increase of the number of DENV-1 infected patients (up to 40% of confirmed cases) was observed compared to 2008 and 2009 when the proportion of DENV-1 was only 10% and DENV-2 was the predominant detected serotype with more than 40% (Figure 7). The number of detected DENV-1 infections further increased in 2011 (77.3%) and became clearly dominant in 2012 (97.5%).

Unfortunately, for most of the first serum samples collected in 2010, RT-PCR results were not available, thus ED3 dot assay results (mED3 single positives) from only 59 second sera could be compared to RT-PCR data (Table 4, 2010). For 2012, from 282 patients RT-PCR data from the first serum sample and mED3 responses from the second serum sample were available (Table 4, 2012). Altogether, 297 (21 + 276) second sera from DENV-1 diagnosed infections from both years were analyzed for serotype-specific antibodies. Only 33% of the diagnosed DENV-1 cases (97/297) showed serotype-specific antibody responses to DENV-1 in the second serum sample (measured as mED3 single positive). The other 200 (8 + 192/297) 2^nd^ sera were positive for serotype-specific antibodies to DENV-2 (33%, 99/297), DENV-3 (14%, 42/297), and DENV-4 (20%, 59/297). None of the 44 second sera from RT-PCR diagnosed DENV-2, DENV-3 or DENV-4 infections showed a matching serotype-specific antibody response. For the 24 second sera from RT-PCR diagnosed DENV-2 infections, 14 were mED3-positive for DENV-1, four second sera were mED3-positive for DENV-3 and six second sera were DENV-4 mED3-positive (Table 4). The lack of mainly DENV-1 serotype-specific antibodies and the emergence of more DENV-2 specific antibody responses in the studied patients might be supportive for the DENV-2-to-DENV-1 shift identified by the RT-PCR surveillance study (Figure 7).

## 4. Discussion

In the last decade, scientific research was focused on domain 3 of the DENV E protein (ED3) and it has been shown that ED3 is a promising vaccine candidate [33] that has reached phase 2 in human vaccine trials [34]. The ED3 domain was expressed in *E. coli* in various ways like fused to trpE [35,36], glutathione-S-transferase [37,38], His_6_ [26,39] or maltose-binding-protein [40]. All these efforts were made to generate ED3 in the most natural conformation possible, and intensive antibody binding studies have identified a correctly folded ED3 when fused to the MBP [41]. Another approach was carried out by Zidane and coworkers [42]. Using a dimeric ED3 hybrid with an N-terminal His_6_-tag and a C-terminal alkaline phosphatase to determine IgM antibodies via MAC-ELISA. They identified a varying specificity for IgM-ED3 recognition of the homotypic serotype for the different DENV serotypes with DENV-1 showing the highest specificity [43]. Antibodies against the histidine-rich antigen of the malaria parasite *Plasmodium* [44] or the glutathione-S-transferase from *Schistosoma japonicum* could lead to undesirable cross-reactions to recombinant proteins tagged with His_6_ or glutathione-S-transferase, especially as malaria and schistosomiasis occur frequently in dengue endemic regions. Based on this knowledge we decided to express ED3 as a fusion protein with MBP. In agreement with all the previously mentioned studies we also achieved a high yield from *E. coli* cultures and a very good overall solubility of our ED3-MBP constructs in contrast to recombinant ED3-His_6_-tagged proteins [26].

The immunogenic and diagnostic potential of ED3 antigens was extensively studied previously. Various ED3 fusion proteins have been shown to induce the production of protective antibodies in mice [45,46] and non-human primates [47]. Recombinant ED3 antigens produced in *Pichia pastoris* [24], insect cells [25] or *E. coli* [26,28,29] were used for several serological assays as well. Due to the presence of DENV cross-reactive and serotype-specific epitopes on ED3 [18,19,20], most of these assays have been used for the detection of DENV antibodies in general but not for the discrimination of antibody responses between the four different DENV serotypes. When applying the ED3 antigens in the non-denatured form we also observed a relatively high proportion of multiple dot-reactivity. This high level of cross-reactivity was not seen with the set of the DENV-1-4 mED3 antigens. Using these modified antigens the number of serotype-specific antibody responses increased significantly with a total number of 679 positive for only one of the DENV serotypes, designated as single-positives in this manuscript. The benefit of ED3 antigens for serotype-specific antibody detection in addition to mED3 was very low (about 1 %). Thus, ED3 itself did not significantly increase the number of serotype-specific responses and therefore can be eliminated as part of the ED3 dot assay in further studies. The impact on antibody analysis using ED3s adds an increase of positives of about 7%. This could be caused by the additional amino acids of the E protein stem part that might stabilize the ED3 domain into a structure that is more close to the natural conformation [48] or might cover cross-reactive epitopes.

The most serotype-specific antibody reactions were obtained with the modified, denatured mED3 antigens. Denaturation of the ED3 antigens possibly destroys conformational epitopes and therefore might reduce the number of epitopes leading to potential cross-reacting antibody responses. This is consistent with the observed decrease in intensity of the mED3 dots and the overall higher amount of negative sera in the mED3 assay. Consequently, we produced truncated versions of ED3 to investigate possible linear epitopes responsible for the serotype-specific reaction. Since the analysis of 24 different N- and C-terminal truncated DENV-1 ED3 constructs revealed no reactivity at all, the explanation of linear epitopes causing a higher serotype-specificity of mED3 was untenable within our technical design. Another possible impact could originate from the MBP fusion protein, although antibody binding studies [41] and structure analysis [49,50] confirmed that ED3 antigens N-terminally fused to MBP and produced in *E.coli* are correctly folded. As biochemical investigations showed that the core of the ED3 domain is unexpectedly rigid and therefore unaffected by wide ranges of pH and temperature changes [51], the MBP fusion might additionally stabilize the antigen core structure. It is also possible that the denaturation process leads to an unknown intermediate or unusual oligomeric state, as recently discovered for DENV-4 ED3 [52]. However, using the ED3 dot assay, DENV-specific antibody responses for only mED3 were observed in 76% of the DENV-positive sera with no cross-reactivity of mED3 to, for example, DENV-negative and/or JEV-positive sera.

Thus far, the confirmatory diagnosis of dengue infection by a serological method still requires a pair of sera, which constitutes a significant limitation. Because of the broad cross-reactivity of IgG antibodies with other flaviviruses, commercial IgG ELISA tests cannot be used to identify the infecting DENV serotype [53]. As a result, in areas where more than one flavivirus is circulating, the pre-existing antibodies and the original antigenic sin phenomenon (B-cells responding to the first infection by synthesis of antibodies with higher affinity than in current infection) [54] make the differential diagnosis of flavivirus infections very difficult [55].

We demonstrated the feasibility of the ED3 dot assay by testing a large cohort of around 1000 sera in two weeks by using 10 μL per sample only. In general, the test gave an overview of the immune status in DENV-positive patients. Since the test correlated specifically with the presence of DENV serotype-specific neutralizing antibodies, such information would enhance the knowledge on DENV-induced immune responses in DENV endemic areas significantly. However, the discordance between RT-PCR-based national surveillance and the detection of the antibody-based serotype with the ED3 dot assay leads to some suggestions. Such discordance between the DENV virus type obtained via RT-PCR and the serotype-specific antibody response obtained with the ED3 dot assay in secondary DENV infections must not be surprising. As mentioned before, the RT-PCR detects the infecting virus directly during the acute phase. In contrast, our study targeted the more complex humoral response. Such an antibody response can reflect a current infection but infection by DENV might also boost humoral responses already primed by previous heterologous DENV-infections, symptomatically or even asymptomatically. As Cambodia is hyper-endemic for DENV infections [56] and all samples originate from the hospital-based national surveillance program, most of the patients in this study are secondary DENV infected. Due to the phenomenon of original antigenic sin it is already known that the antibody response to a secondary DENV infection does not always target the infecting virus but the virus of the primary infection [57]. In this context, it would be interesting to monitor patients over a longer period to find out if they will develop a serotype response against a second or third infection or if the priming by the first infection still remains as the dominant DENV response. The discordance between virus type obtained via RT-PCR and the serotype-specific antibody response obtained with the ED3 dot assay in primary infections could be due to the fact that the early humoral immune response after a primary infection is dominated by cross-reacting antibodies. Additionally, it is known that the virus type solely determined by the virus genome and the specificity determined by the individual’s immune response to a certain virus antigenic structure do not always match [58].

The ED3 dot assay has shown that the DENV-1 serotype-specific responses are missing in most of the DENV-1 infected symptomatic patients. The same was observed for all other DENV infections although the number of patients with this characteristic was much lower. Thus, it can be speculated, a missing or low serotype-specific response could be a risk factor for a single individual to become infected by a different DENV not matching the detected serotype-specific antibody response, especially as all analyzed Cambodian samples were exclusively from hospitalized patients. The ED3 dot assay would be a well-suited tool for monitoring such antibody gaps to Dengue.

On the other hand, such serological diagnostic tools that would enable the identification of the DENV serotype-specific antibody response in endemic areas could be of great interest. Tests like the ED3 dot assay would allow an easy epidemiological surveillance, outbreak investigations and the monitoring of vaccine efficacy trials [11] as neutralization assays are extremely labor-intensive, expensive, and can only be performed in a few reference laboratories. The ED3 dot assay would for example allow the monitoring of the anti-DENV immune status in communities before and after vaccination trials. 

Although WHO reference strains are defined and are distributed between laboratories, technical deviations have been described [14,15]. However, the ED3 dot assay was produced using sequences from reference strains. The resulting serotype-specific antibody responses identified with the ED3 dot assay were compared to the results of a neutralization test, using the reference DENV strains, and an excellent concordance of the results was observed especially for DENV-1 and DENV-2. Equal agreement was observed between the ED3 dot assay and the ED3 ELISA using the same reference antigens. However, it might be necessary to adapt the constructs to recent circulating DENV strains from certain regions as this could lead to higher specificity and sensitivity of the test. A poor match of reference strain and recently present DENV could explain the rather low reactivity of sera against ED3 from DENV-3 in the dot assay and the low neutralization titer (Figure 4C), as both assays used the DENV-3 reference strain H87. Especially for DENV-3 a high antigenic diversity within the serotype was documented [59].

Two other ELISA techniques developed for serotype analysis are described throughout the literature. Firstly, Libraty et al. [29] also performed ELISAs using recombinant ED3 antigens, but evaluated this test only with sera of primary infected children. Secondly, an immune complex binding (ICB) ELISA also using ED3 antigens has shown to detect serotype-specific reactions essentially in selected primary infections [28]. However, the number of tested samples was low in both studies and therefore might be not representative. In our study, we used sera obtained from patients with primary as well as multiple sequential infections (noted as secondary infections) as determined by standard diagnostic HI tests performed on paired sera. Additional to the sera displaying a single serotype reaction, 19 % of the sera tested in the Cambodian survey were reacting simultaneously against 2 different serotypes. These findings are comparable with the reported 21% of double-positives with the ICB ELISA [28].

The here presented ED3 ELISA, alike the ED3 dot assay, can also be performed easily and does not require a safety laboratory as the classical DENV neutralization assay does. Nevertheless, this ELISA requires several dilution steps (in triplicate) to distinguish between cross-reactive and serotype-specific reactions. However, the ED3 ELISA would be best suited for single patient diagnostics and not for large retrospective studies. A benefit of the ED3 dot assay is the requirement of only one serum dilution, thus making the procedure less complex compared to ELISA and neutralization tests. Another advantage of the ED3 dot assay is that dried test strips can be stored at room temperature without loss of sensitivity (tested for 55 days, data not shown), which simplifies the storage, handling and shipping process significantly. The ED3 dot assay can be used for higher throughput analysis as a team of two trained technicians were able to perform the analysis of around 1000 sera in less than two weeks.

In summary, the application of the DENV ED3 dot assay allows an uncomplicated screening for the dominant DENV-specific humoral response in endemic areas in patients but also in non-infected individuals or vaccinees. The knowledge of the predominant DENV serotype-specific antibody in a population would be very valuable as, amongst other reasons, serotype replacements contribute to peak epidemics [60]. Therefore, the application of mED3 antigens in surveillance studies in combination with surveillance of the active circulating virus can lead to enhanced preparedness and quicker responds to DENV outbreaks. The ED3 dot assay showed the potential to obtain such valuable serotype-specific data in epidemic regions and large cohorts.

## Figures and Tables

**Figure 1 viruses-11-00304-f001:**
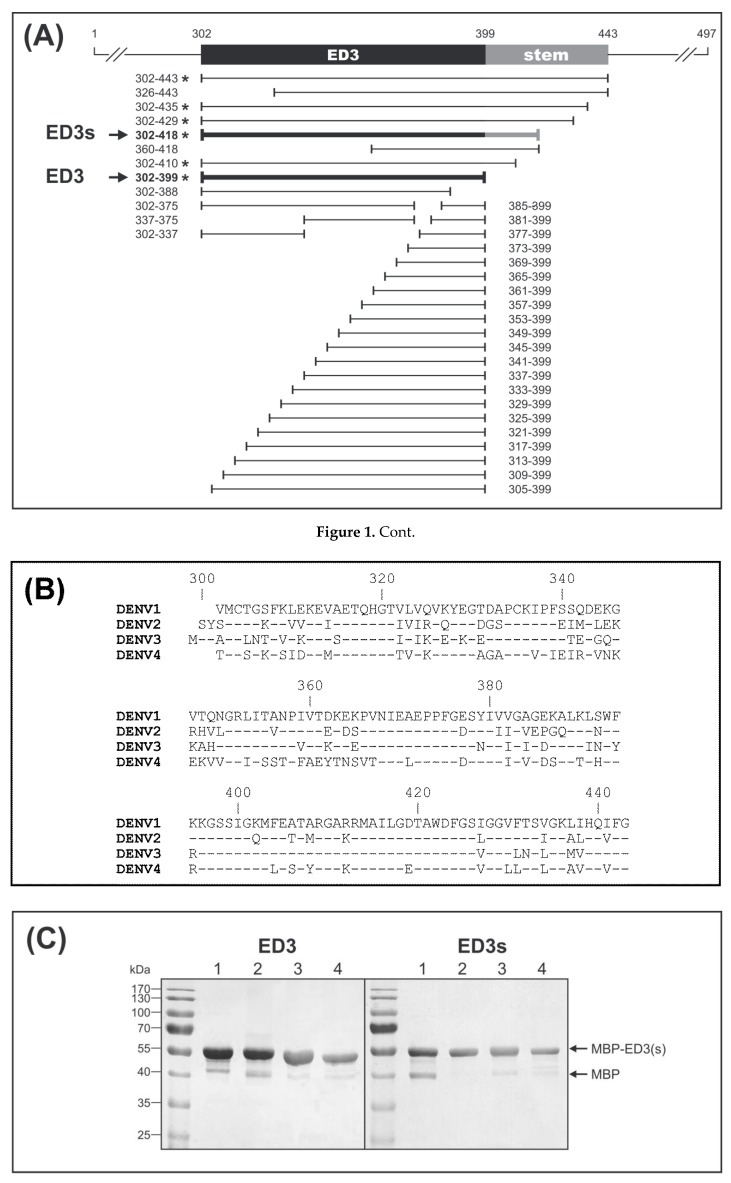
Antigens for the ED3 dot assay. (**A**) DENV-1 ED3 constructs tested for serum reactivity. All ED3 constructs were expressed in *Escherichia coli* BL21 (DE3) fused to MBP. Antigens that reacted positively with DENV-1 sera were marked by an asterisk. Antigens used in the ED3 ELISA and ED3 dot assay: ED3 (aa 302–399) and ED3s (aa 302–418) marked by an arrow. (**B**) ED3-stem sequences. DENV-1 West Pac (GenBank: U88535), DENV-2 TH/BID V3357/1964 (GenBank: GQ868591), DENV-3 H87 (GenBank: M93130), DENV-4 PH/BID V3361/1956 (GenBank: GQ868594). (**C**) Antigens purified by amylose affinity and size exclusion chromatography. Purified MBP-ED3-fusion proteins were analyzed on a silver-stained 10% SDS-polyacrylamide gel. Lane numbers indicate DENV-1-4. (**D**) Design of the ED3 dot assay test strip. Numbers indicate DENV-1-4. mED3, denatured form of ED3. DENV, Dengue virus; WNV, West Nile virus; JEV, Japanese encephalitis virus; TBEV, Tick-borne encephalitis virus; MBP, maltose binding protein.

**Figure 2 viruses-11-00304-f002:**
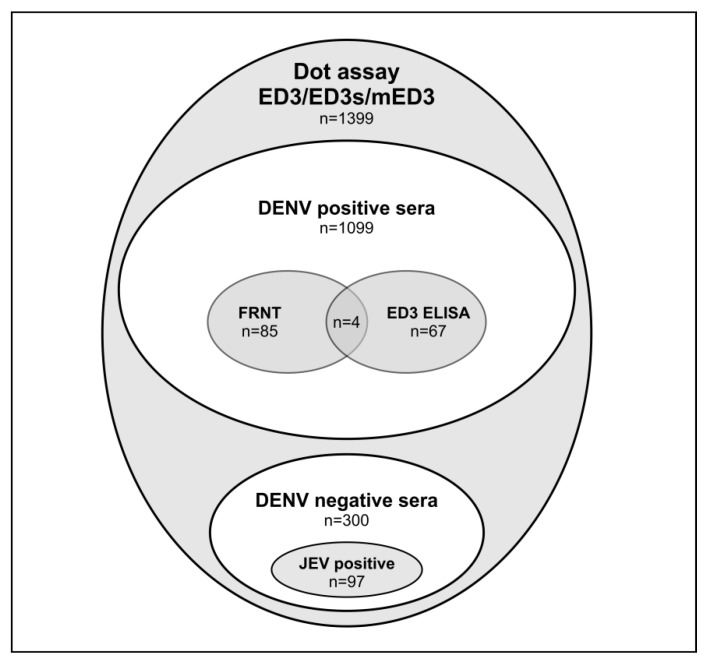
Sera used within this study. All 1399 sera were tested with the dot assay using recombinant ED3, ED3s and mED3 antigens of DENV-1-4. Samples included 1099 2^nd^ sera from confirmed DENV cases and 300 DENV-negative sera, the latter included 97 sera from JEV patients. Dot assay results of 67 sera were compared to ED3 ELISA results. Dot assay results of 85 sera were compared to foci reduction neutralization test (FRNT) performed to detect serotype-specific antibodies to one of the DENVs. ELISA, enzyme-linked immunosorbent assay.

**Figure 3 viruses-11-00304-f003:**
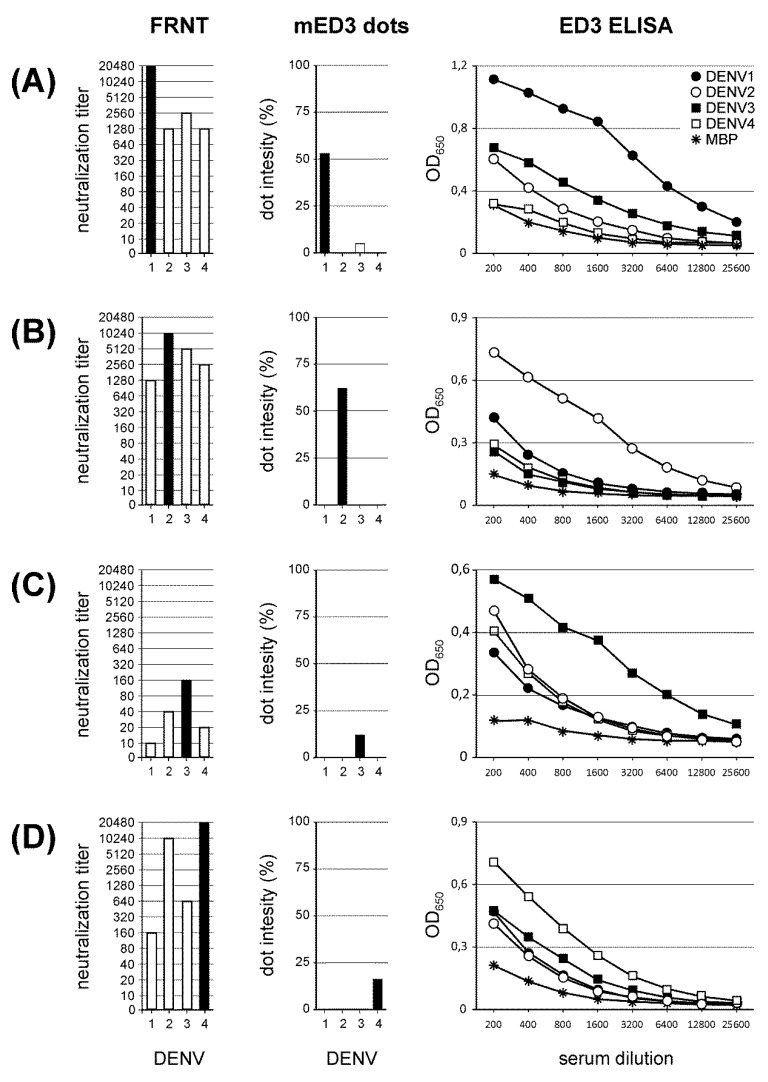
Comparison of FRNT, mED3 dot intensity and ED3 ELISA. Assay data are exemplarily shown for the detection of DENV-1-4 serotype-specific antibodies (**A**–**D**, respectively). The antibody specificity determined by FRNT is indicated by the highest FRNT_90_ titer (black bars, x-axis; numbers indicate DENV-1-4). The specificity detected by mED3 is indicated as the highest intensity obtained, based on data observed by the ED3 dot assay (black bars, x-axis; numbers indicate DENV-1-4). Serotype-specific antibody responses analyzed by ED3 ELISA are indicated by the highest OD_650_ value through all serum dilutions. ED3 antigens for DENV-1, black circles; DENV-2, white circles; DENV-3, black squares; DENV-4, white squares and MBP as a control, asterisks.

**Figure 4 viruses-11-00304-f004:**
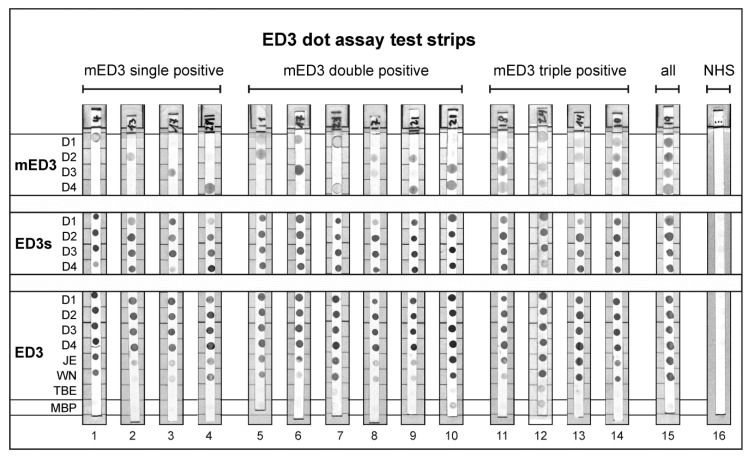
ED3 dot assay test strips. Exemplarily shown are test strips results for all combinations of mED3 positivity. Sera used for the assay are described in Table 3. NHS, normal human serum negative for DENV and JEV antibodies.

**Figure 5 viruses-11-00304-f005:**
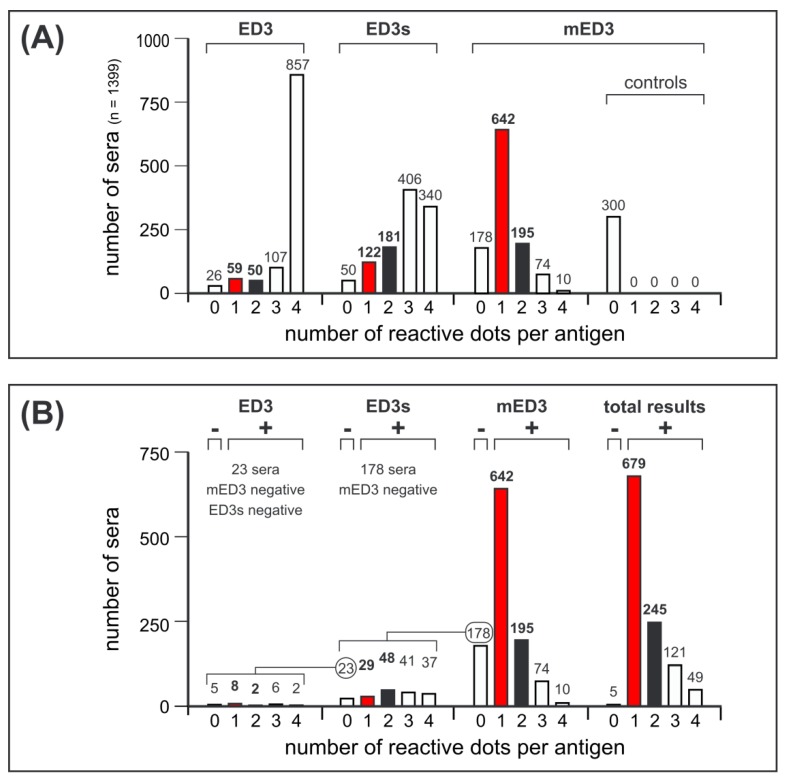
Results of the ED3 dot assay. Reactivity of sera to the DENV-1-4 antigens, ED3, ED3s and mED3. Red bars indicate sera with single-dot reactivity to only one of the four DENV antigens within the respective antigen panel; black bars indicate double-reacting serum samples. (**A**) Number of sera reactive to each set of DENV-1-4 antigens. x axis: 0 = not reactive, 1 = single dot positive, 2 = double dot positive, 3 = triple dot positive, 4 = quadruple dot positive; y axis: number of sera. (**B**) Number of sera (*n* = 1099) with single, double, triple and quadruple reactivity tested by the ED3 dot assay. X axis: 0 = not reactive, 1 = single dot positive, 2 = double dot positive, 3 = triple dot positive, 4 = quadruple dot positive; y axis: number of sera. (**C**) Total results of the ED3 dot assay related to DENV-1-4.

**Figure 6 viruses-11-00304-f006:**
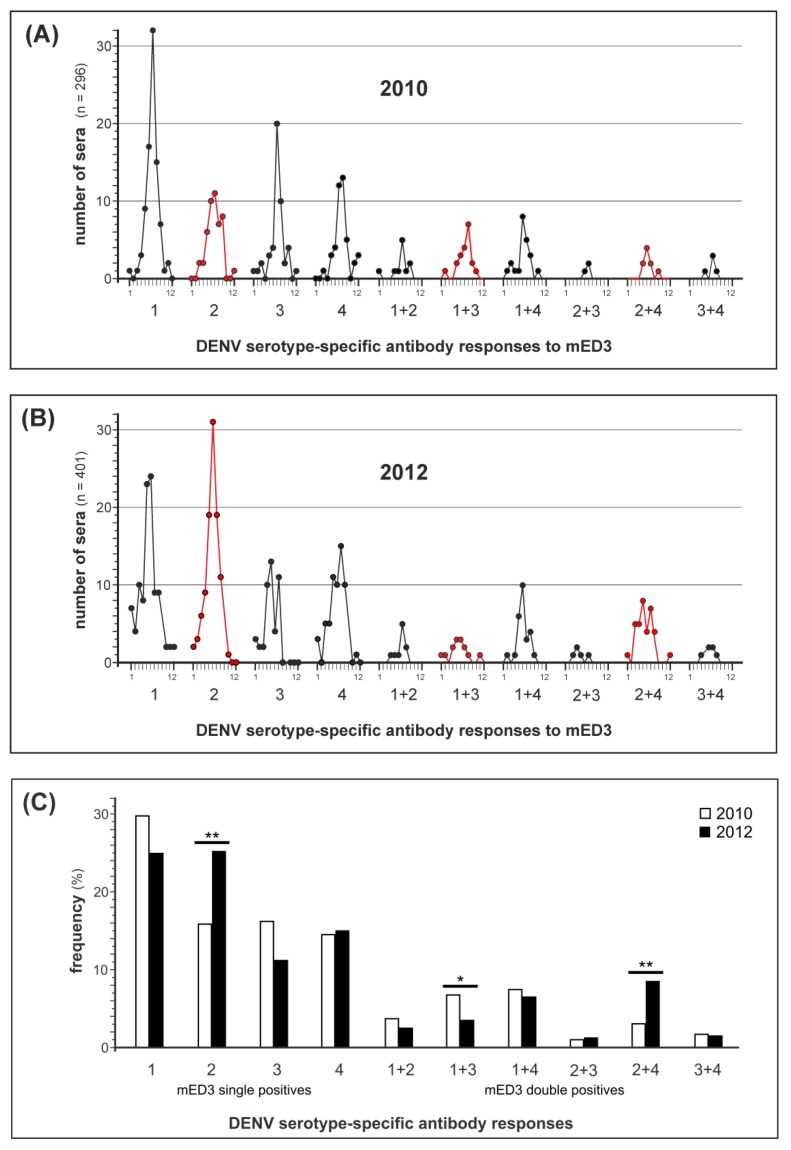
Results of the retrospective seroprevalence study. (**A**) ED3 dot assay results in 2010 given per month (1–12). Sera were all tested mED3 dot single- (*n* = 226) or double-positive (*n* = 70) and are a subset of the sera (*n* = 296) as shown in Figure 5B (*n* = 924). (**B**) ED3 dot assay results in 2012. Sera were tested single- (*n* = 306) or double-positive (*n* = 95) and are a subset (*n* = 401) of sera shown in Figure 5B (*n* = 924). Red circles, sera positive for DENV-2/1 + 3/2 + 4 that show significant changes in serum numbers. (**C**) Frequency of DENV serotype-specific antibody responses in 2010 and 2012. Frequencies were calculated based on the numbers of positive sera as shown above in Figure 6A,B. White bars, frequency of serotype-specific antibody responses detected in 2010. Black bars, frequencies detected in 2012. Significant differences were calculated using a 2-tailed z-test and are labeled by asterisks (* *p* = 0.047, ** *p* = 0.003).

**Figure 7 viruses-11-00304-f007:**
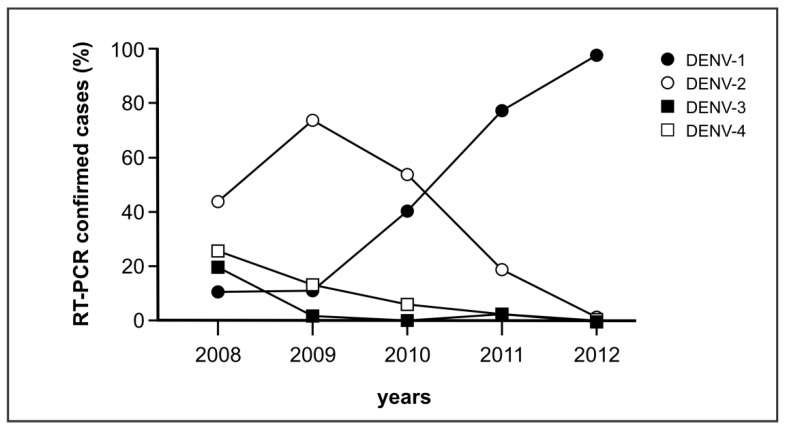
Frequency of DENV-1-4 in Cambodia from 2008 to 2012 analyzed by RT-PCR. Data are from the IPC DENV surveillance study [32].

**Table 1 viruses-11-00304-t001:** Comparison of test results between ED3 dot assay and Foci Reduction Neutralization Test (FRNT).

Serotype Specificity by ED3 Dot Assay	No. Sera Tested	Serotype Identified by FRNT				Matching Results (%)	
		DENV-1	DENV-2	DENV-3	DENV-4		
DENV-1	31	29	2	0	0	29	(93.5)
DENV-2	25	0	25	0	0	25	(100)
DENV-3	19	0	3	15	1	15	(79)
DENV-4	10	0	4	0	6	6	(60)
total	85					75	(88)

**Table 2 viruses-11-00304-t002:** Comparison of test results between ED3 dot assay and ED3 Enzyme-linked Immunosorbent Assay (ELISA).

Serotype Specificity by ED3 Dot Assay	No. Sera Tested	Serotype Identified by ED3 ELISA				Matching Results (%)	
		DENV-1	DENV-2	DENV-3	DENV-4		
DENV-1	25	23	0	2	0	23	(92)
DENV-2	22	0	22	0	0	22	(100)
DENV-3	12	2	0	10	0	10	(83)
DENV-4	8	0	1	0	7	7	(87.5)
total	67					62	(92.5)

**Table 3 viruses-11-00304-t003:** Characteristics of patient sera used for the development of test strips as shown in Figure 4.

Serum mED3 positive for DENV	RT-PCR Serotype	Days After Onset of Fever	IgM MAC-ELISA		HIA		
			DENV	JEV	DENV-2	DENV-3	JEV
1	DENV-1	9	positive	positive	0	80	0
2	DENV-1	8	positive	negative	10,240	5120	2560
3	DENV-1	7	positive	positive	10,240	20,480	20,480
4	DENV-2	8	positive	positive	1280	1280	1280
1 + 2	DENV-1	8	positive	negative	2560	2560	1280
1 + 3	DENV-1	8	positive	positive	5120	10,240	5120
1 + 4	DENV-1	7	n.d.	n.d.	160	320	320
2 + 3	DENV-1	8	positive	positive	20,480	20,480	20,480
2 + 4	DENV-1	7	positive	positive	10,240	5120	2560
3 + 4	DENV-1	9	positive	positive	20,480	20,480	10,240
2 + 3 + 4	DENV-1	6	positive	positive	5120	5120	2560
1 + 3 + 4	DENV-1	7	nd	nd	80	40	80
1 + 2 + 4	DENV-1	7	positive	positive	10,240	20,480	5120
1 + 2 + 3	DENV-1	7	positive	positive	10,240	10,240	1280
1 + 2 + 3 + 4	DENV-1	7	positive	positive	20,480	20,480	20,480
NHS	negative	-	negative	negative	negative	negative	negative

**Table 4 viruses-11-00304-t004:** Comparison of mED3 antibody responses and DENV serotypes detected by RT-PCR.

**2010**							
**Serotype by RT-PCR**	**No. Sera Tested**	**Serotype-Specific Antibody Response Identified by mED3 ***				**Matching Results (%)**	
		**DENV-1**	**DENV-2**	**DENV-3**	**DENV-4**		
DENV-1	21	13	4	1	3	13	(61,9)
DENV-2	19	12	0	3	4	0	(0)
DENV-3	15	9	3	0	3	0	(0)
DENV-4	4	3	1	0	0	0	(0)
total	59	37	8	4	10	13	(22,0)
**2012**							
**Serotype by RT-PCR**	**No. Sera Tested**	**Serotype-Specific Antibody Response Identified by mED3 ***				**Matching Results (%)**	
		**DENV-1**	**DENV-2**	**DENV-3**	**DENV-4**		
DENV-1	276	84	95	41	56	84	(30,4)
DENV-2	5	2	0	1	2	0	(0)
DENV-3	0	0	0	0	0	0	(0)
DENV-4	1	0	1	0	0	0	(0)
total	282	86	96	42	58	84	(29,8)

* The avarage time frame between onset of fever and collection of serum samples was 7,3 ± 1,4 days.

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
