# Peer review of "The Dengue ED3 Dot Assay, a Novel Serological Test for the Detection of Denguevirus Type-Specific Antibodies and Its Application in a Retrospective Seroprevalence Study"

_viruses, 2019, doi:10.3390/v11040304_

Round 1

Reviewer 1 Report

In the manuscript Auerswald et al the authors describe the expression of Dengue virus (DENV) E protein domain 3 (ED3) for all four DENV serotypes (1-4) fused to maltose binding protein (MBT) in E. coli. After expression of several truncated ED3 domains including the ED3 stem region and testing the expressed proteins for DENV1 sera reactivity, the authors decided on two proteins (ED3 and ED3s) for further serologic analyses. Besides including nature ED3/s proteins in their studies the purified proteins were also denatured and compared in their reactivity against non-denatured ED3/s. Furthermore, the respective ED3 domains of other flaviviruses (WNV, JEV, TBE) were included. Using these proteins, the reactivity and specificity of the different proteins were tested in a dot assay with the aim to determine the most suitable antigen and to analyse serum samples from Cambodia sampled in 2010 and 2012. The manuscript comprises a detailed and very thorough analysis of the suitability of the established dot ELISA.

The authors should consider the following points to improve the manuscript:

Abstract/Lane 23: it is stated that different MBP-antigens were dotted included WNV, JEV etc. From this sentence it is not clear whether all four DENV antigens were also included in the assay. This question especially comes up when the authors later in the abstract mention that the test is useful for DENV1 and DENV2 serotype specific antibody detection – what about the other two DENVs? They have been tested but with no good results. If this is not mentioned than it is hard to understand what was done. Thus the abstract should mention that DENV1-4 is included in the test.

Lane 44: it is stated that the cross-reactive antibodies produced against a heterologous serotype are short-lived. Are the cross-reactive antibodies really produced against a heterologous virus? Or are the antibodies produced against the virus from the first infection cross-reactive against the virus (different serotype) from the second infection? Rephrase?

Lane 183 ff: This seems to be a house made ELISA using the same antigens as used in the dot ELISA presented in the paper. Has this ELISA ever been compared to a commercially available and validated ELISA?

Lanes 274ff/ Section 3.2

Overall, this section seems to overlap with section 3.4, which also describes the comparison of the different antigen types. As the manuscript is anyway quite lengthy (25 pages!) these two sections should be combined.

Furthermore, and most importantly  –  in Figure 2 it is not clear how the sera used were characterized beforehand by other assays (positive against which DENV? And which assay was used?).  Hence, above each stripe it must be listed by which assay the sera has been analysed before and what was the result of this assay.

Also – what does it mean if the sera reacts with different antigens. Is this considered cross-reactivity or double infection? To better judge this it would also be good to know what was determined for the other assays using other tests.

Minor: lane 279 – shouldn’t it be mED3s positives instead of mED3 positives?

Minor: lane 281: quadruple instead of quadrupole?

Figure 4: for better comparison: same scale should be used in the corresponding y axis. As for Fig. 4C the NT titer the value is quite low, an inlay as shown could be inserted next to the figure with the scale up to 20480.

Also: how do the authors explain that comparing Fig 4C to 4D the mED3 dot signals are almost the same whereas the NT titers differ about 100 fold?

Section Lines 360-371: The conclusions drawn of Figure 5B are difficult to follow. Where does the improvement rate of serotype-specificity of 7% for ED3s is concluded from? And where the 1% for ED3?  Where in the figure are the 300 DENV-negative sera depicted that are mentioned in the paragraph?

Figure 5C: What does it mean if the sera are double positive: double infection or cross reactivity. Is there a certain threshold value (e.g. more than four fold higher) that can be determined to differentiate between cross-reactivity and double infection?

Minor: lane 367: Figure 5C instead of 4C?

Figure 6:

In A the two highest peaks were chosen to point out (DEN-1/-3/-1+3).

Why were in Figure B Den-2/-4/2+4 chosen? The DEN-2 and DEN-4 peaks are not the two highest once. DEN-1 is higher than Den-4. What is the ratio behind pointing out the sera positive for DENV-2/-4/2+4 in Figure B? This needs better explanation.

Table3:

The authors compare here the mED3 antibody responses and DENV serotypes detected by RT-PCR. Are the samples used from the same serum aliquot? If so, this would mean that the sera are from an early viraemic phase of infection. Usually at that time antibodies are not produced yet and the comparison would not make much sense and the Table can be omitted. Or are the sera used for the dot ELISA follow-up samples from the RT-PCR tested samples? If so, what is the time frame between 1st and 2nd sera?

Author Response

Response to comments of Reviewer 1

Reviewer:   In the manuscript Auerswald et al the authors describe the expression of Dengue virus (DENV) E protein domain 3 (ED3) for all four DENV serotypes (1-4) fused to maltose binding protein (MBT) in E. coli. After expression of several truncated ED3 domains including the ED3 stem region and testing the expressed proteins for DENV1 sera reactivity, the authors decided on two proteins (ED3 and ED3s) for further serologic analyses. Besides including nature ED3/s proteins in their studies the purified proteins were also denatured and compared in their reactivity against non-denatured ED3/s. Furthermore, the respective ED3 domains of other flaviviruses (WNV, JEV, TBE) were included. Using these proteins, the reactivity and specificity of the different proteins were tested in a dot assay with the aim to determine the most suitable antigen and to analyse serum samples from Cambodia sampled in 2010 and 2012. The manuscript comprises a detailed and very thorough analysis of the suitability of the established dot ELISA.

                   The authors should consider the following points to improve the manuscript:

Abstract/Lane 23: it is stated that different MBP-antigens were dotted included WNV, JEV etc. From this sentence it is not clear whether all four DENV antigens were also included in the assay. This question especially comes up when the authors later in the abstract mention that the test is useful for DENV1 and DENV2 serotype specific antibody detection – what about the other two DENVs? They have been tested but with no good results. If this is not mentioned than it is hard to understand what was done. Thus the abstract should mention that DENV1-4 is included in the test.

Response:   The abstract has been changed accordingly.

Reviewer:   Lane 44: it is stated that the cross-reactive antibodies produced against a heterologous serotype are short-lived. Are the cross-reactive antibodies really produced against a heterologous virus? Or are the antibodies produced against the virus from the first infection cross-reactive against the virus (different serotype) from the second infection? Rephrase?

Response:   The sentence was changed to: ….. However, this immunity does not provide long-time protection against infection with another serotype as the cross reactive antibodies produced from the first infection  are short-lived and have poorly or non-neutralizing activity against other serotypes [4,5].

Reviewer:   Lane 183 ff: This seems to be a house made ELISA using the same antigens as used in the dot ELISA presented in the paper. Has this ELISA ever been compared to a commercially available and validated ELISA?

Response:   Both the ED3 ELISA and the ED3 dot assay are house-made tests. In both assays we are using the ED3 antigens originated from the same DENV strains that were used in our FRNTs. We compared the results from dot assay and ELISA with the serotype obtained by the NT as this is the only serological standard assay that can determine the serotype. In contrast, commercially available ELISAs detect the presence of antibodies but not their virus-specificity.           

Reviewer:   Lanes 274ff/ Section 3.2

                   Overall, this section seems to overlap with section 3.4, which also describes the comparison of the different antigen types. As the manuscript is anyway quite lengthy (25 pages!) these two sections should be combined.

            Response:   3.2 has now been combined with 3.4.

Reviewer:   Furthermore, and most importantly  –  in Figure 2 it is not clear how the sera used were characterized beforehand by other assays (positive against which DENV? And which assay was used?).  Hence, above each stripe it must be listed by which assay the sera has been analysed before and what was the result of this assay.

Response:  The sera used in this study where diagnosed for DENV depending on their origin (as described in section 2.3). On the test strips of Figure 4 all sera are from Cambodia. They were diagnosed for the presence of DENV via RT-PCR and DENV-specific antibodies by IgM MAC_ELISA and HIA.

                   We have added a table (Tab. 3) with the diagnostic data of the sera used in Figure 4.

Reviewer:   Also – what does it mean if the sera react with different antigens? Is this considered cross-reactivity or double infection? To better judge this it would also be good to know what was determined for the other assays using other tests.

Response: Our ED3 dot assay is very similar to the antibody typing published by Dejnirattisai et al. (ref. 26). They used SDS-PAGE and western blot and a dot blot assay to study specificity of prM, NS1 and E directed antibodies. They also classified sera positive and negative to a DENV by Yes and No decisions as can be seen in their supplementary data (Tab S3 in ref 26). What is called fully cross reactive by Dejnirattisai et al. is quadruple-positive in our paper, partial-cross reactivity is double- or triple-positive in our paper and serotype-specific is single mED3 positive in our paper. Again, single-positive to mED3 by our ED3 dot assay matched our NT results for the detection of the highest titer to one of the DENVs. Therefore, we are sure that we are technically in agreement with these antibody typing studies.

                   Since in NT studies antibodies are binding to virion associated E and ED3 we believe it is not possible to decide between cross-reactivity or double infection by the denatured mED3 antigens. We would suggest that when people are single-positive to DENV-1 mED3 they will be at risk for the other three DENVs. The ED3 dot assay, it's more for looking into the future than looking into the history of infection.

                   Our hypothesis is, that people develop a DENV-specific response by natural infection, we call it priming. This response can be boosted by a second infection or a vaccine leading to higher titers and better neutralization. In vaccine studies this is documented as lower number of hospitalized DENV cases. Here the question is, will a second infection only boost the primed response or can it start a new specific one. The actual problem with the tetravalent vaccine when administered in DENV naïve people points to the prime and booster theory. The vaccine response is boosted by natural infection but does not help to control the specific DENV strains present in that area.

                   Alltogether, reaction to several DENV ED3 antigens as observed with our dot assay can be due to (i) cross-reactivity of the antibody response short after a primary infection, (ii) cross-reactivity long-term after a secondary infection, or (iii) double infection . Neither our study design (with the samples that we could access) nor the dot assay itself are able to decipher the reason for the observed double or multiple reactivity.

Reviewer:Minor: lane 279 – shouldn’t it be mED3s positives instead of mED3 positives?

                   Minor: lane 281: quadruple instead of quadrupole?

                   Minor: lane 367: Figure 5C instead of 4C?

 Response: lane 279, yes that’s incorrect. Specific reactions were observed for the mED3 set of antigen not with the two sets for ED3s and ED3. It is now written more precisely:

                   Reactivity of DENV-positive sera against ED3 and ED3s antigens showed under the experimental conditions no serotype-specific reactivity (Figure 4, serum dilution 1:100). Most of the ED3s or ED3 antigens reacted to sera and especially all the ED3 antigens were highly reactive. ED3 for JEV and WNV also showed strong responses whereas sera were negative to TBEV. One of the sera showed a weak response to the MBP control but clearly distinguished between the four mED3 antigens (double mED3 positive for DENV-3, -4, negative for DENV-1, -2). In contrast to ED3 and ED3s, the mED3 antigen panel reacted more specific and we observed single, double, triple positive dot and also clearly negative dot results. Also quadruple responses were detected by mED3 antigens (Figue 4, all). These sera reacted also strongly to the other 8 DENV-1-4 ED3 and ED3s antigens. Thus, in these overall reactive sera no serotype-specific response could be detected by mED3, In Table 3 direct and indirect diagnostic parameters of the sera used in Figure 4 are shown.

                   lane 281: Yes, quadruple

                   lane 376: Yes, 5C

Reviewer:   Figure 4: for better comparison: same scale should be used in the corresponding y axis. As for Fig. 4C the NT titer the value is quite low, an inlay as shown could be inserted next to the figure with the scale up to 20480.

                   Also: how do the authors explain that comparing Fig 4C to 4D the mED3 dot signals are almost the same whereas the NT titers differ about 100 fold?

Response: The y axis for the NT titers is now divided into equal steps for each dilution. (figure 3 in the new version).

                   As mentioned before , our ED3 dot assay is very similar to the antibody typing published by Dejnirattisai et al. (ref. 26). They also made Yes/No decisions when typing the human monoclonal antibody responses. As antibodies in NT bind to the virion associated proteins we think these titers cannot be compared to titers of antibodies directed against denatured recombinant antigens. The mED3 single dot response just points out what is the highest titer in NT. The dot intensity is not directly correlated to the titer of neutralizing antibodies as the ED3 dot assay measures antibodies based on their specific binding to certain recombinant antigens, and the NT detects antibodies with a neutralizing function to certain viruses. To our knowledge that is in agreement with the published data.

Reviewer:   Section Lines 360-371: The conclusions drawn of Figure 5B are difficult to follow. Where does the improvement rate of serotype-specificity of 7% for ED3s is concluded from? And where the 1% for ED3?  Where in the figure are the 300 DENV-negative sera depicted that are mentioned in the paragraph?

Response: In the mED3 negative group 7% reacted to one or two ED3s antigens

                   changed in the text: …7% (Figure 5B, 29 + 48/1099 sera)…

                   In the mED3 and ED3s negative group 1% reacted to one or two ED3 antigens

                   changed in the text:  …1% (Figure 5B, 8 + 2/1099 sera)….

                   300 DENV-negative sera are in Figure 5A. The sentence was added into the 5A paragraph.

Reviewer:   Figure 5C: What does it mean if the sera are double positive: double infection or cross reactivity. Is there a certain threshold value (e.g. more than four fold higher) that can be determined to differentiate between cross-reactivity and double infection?

Response:  As mentioned in a comment above, the observed double reactivity in the dot assay could be due to cross-reactivity of the antibody response short after a primary infection, or long-term after a secondary infection, or due to a double infection. In neutralization tests (NTs) performed with paired sera (collected in acute and convalescent phase) cross-reactivity can often be ruled out by identifying the serotype/virus with the highest increase of the NT titer. Based on the WHO criteria for the interpretation of flavivirus NT data, a 4-fold increase indicates the most recent infecting virus. However, the authors are not aware of any NT studies that explored the immune response after double-infection, despite several reports of virologically confirmed (via PCR) double infections.

Reviewer:   Figure 6:   In A the two highest peaks were chosen to point out (DEN-1/-3/-1+3).

Why were in Figure B Den-2/-4/2+4 chosen? The DEN-2 and DEN-4 peaks are not the two highest once. DEN-1 is higher than Den-4. What is the ratio behind pointing out the sera positive for DENV-2/-4/2+4 in Figure B? This needs better explanation.

Response: In red, the set of sera with significant changes as shown in C are stained red now. The Figure has been changed accordingly.

Reviewer:   Table3: The authors compare here the mED3 antibody responses and DENV serotypes detected by RT-PCR. Are the samples used from the same serum aliquot? If so, this would mean that the sera are from an early viraemic phase of infection. Usually at that time antibodies are not produced yet and the comparison would not make much sense and the Table can be omitted. Or are the sera used for the dot ELISA follow-up samples from the RT-PCR tested samples? If so, what is the time frame between 1st and 2nd sera?

Response:  We thank the reviewer for the useful comment. The initial DENV diagnosis with the Cambodian sera has been performed with an acute phase serum sample (1-3 days after onset of symptoms) analyzed by RT-PCR, HIA and IgM ELISA. The dot assay was performed with follow-up samples from the early convalescent phase (7±2 days after onset of symptoms).

                   This is the new version:

Unfortunately, from paired serum samples collected in 2010, RT-PCR results from the 1st  serum were not available, thus ED3 dot assay results (mED3 single positives) from only 59 2nd sera could be compared to RT-PCR data (Table 4, 2010). For 2012, from 282 patients RT-PCR data from the 1st serum sample and mED3 responses from the 2nd serum sample were available (Table 4, 2012). Altogether, 297 (21+276) sera from DENV-1 diagnosed infections were analyzed for serotype-specific antibodies. Only 33% of the diagnosed DENV-1 cases (97/297) showed serotype-specific antibody responses to DENV‑1 in the 2nd serum sample (measured as mED3 single positive). The other 200 sera (8+192/297) were positive for DENV-2 (33%, 99/297), DENV-3 (14%, 42/297), and DENV-4 (20%, 59/297). None of the 44 2nd sera from RT-PCR diagnosed DENV-2, DENV-3 or DENV‑4 infections showed a matching serotype-specific antibody response. For the 24 2nd sera from RT-PCR diagnosed DENV-2 infections, 14 were mED3-positive for DENV-1, four positive for DENV-3 and six positive for DENV-4 (Table 4). The lack of mainly DENV-1 serotype-specific antibodies and the emergence of more DENV-2 specific antibody responses in the studied patients might be supportive for the DENV-2-to-DENV-1 shift identified by the RT-PCR surveillance study (Figure 7).

Reviewer 2 Report

The authors have developed a quick and cheap method to test the serotype from DENV-infected patients. They used it to retrospectively analyse patient samples from existing cohorts in Cambodia. Overall the manuscript is not well written and not easily understandable. The discussion helps the readers to better understand the study achievements but this is rather insufficient to get the manuscript published in the current format. The manuscript needs extensive reformatting to get improved. The authors also need to add robust arguments regarding the major points raised by the reviewer.

Major points:

1) Regarding the existing literature (e.g. neutralising antibodies that recognise discontinuous epitope), it is rather surprising that denatured peptides give a better specificity that their respective folded forms. Authors must confirm that this issue is not due to a production artefact.

- Disulfide bond formation occurs in the bacterial periplasm of Gram-negative bacteria and enables the proper folding of endogenous and exogenous expressed proteins/peptides. E.coli BL21 (DE3) have not been optimized for disulfide bond formation. Authors must confirm their assay with peptides expressed in bacteria modified to allow the stable formation of disulphide bonds in their cytoplasm.

- Overexpressed peptides have not been optimized for bacterial expression and their might be some issues regarding rare codons. Authors must verify that their expression system assures the proper production of their peptides of interest.

As the study rational only relies on the ED3 dot assay, authors must clarify these two latter points to access publication in Viruses.

2) English writing is poor and must be improved. This makes the manuscript track not easy to follow and the understanding not optimal. For example, the following section is absolutely not understandable. What do the authors mean? This is an important section of the manuscript to understand the following sample analysis and it really lacks of pedagogy.

« Reactivity of DENV-positive sera against ED3 antigens showed under the experimental conditions (Figure 2, serum dilution 1:100) that in the presented set of mED3 positives, no serotype-specific reactivity was observed. In contrast, the mED3 antigen panel reacted more specific and we observed single, double, triple and quadrupole serotype-specific antibody responses. »

Minor points:

1) « Since the majority of the antibodies are not serotype-specific [26], ED3 antigens are rarely used for the detection of serotype-specific anti-DENV-antibodies [27,28]. Altogether, the specificity of ED3-derived antigens for the determination of DENV serotypes and their application in a serological test seems to be controversial and needs further clarification. »

Authors must temperate their assumption that ED3 epitopes have not been related to DENV-specific antibodies as a large body of literature is showing that ED3 induces sero-specific rather than cross-reactive antibodies.

2) Figure 1B- highlighting differences in the DENV sequences would be helpful for the readers.

3) Typing errors occur along the manuscript. Make sure that appropriate changes will be made in the revised manuscript.

4) Please update the authors details (Buchy P).

Author Response

Response to comments of Reviewer 2

The authors have developed a quick and cheap method to test the serotype from DENV-infected patients. They used it to retrospectively analyse patient samples from existing cohorts in Cambodia. Overall the manuscript is not well written and not easily understandable. The discussion helps the readers to better understand the study achievements but this is rather insufficient to get the manuscript published in the current format. The manuscript needs extensive reformatting to get improved. The authors also need to add robust arguments regarding the major points raised by the reviewer.

Major points:

Reviewer:   1) Regarding the existing literature (e.g. neutralising antibodies that recognise discontinuous epitope), it is rather surprising that denatured peptides give a better specificity that their respective folded forms. Authors must confirm that this issue is not due to a production artefact.

- Disulfide bond formation occurs in the bacterial periplasm of Gram-negative bacteria and enables the proper folding of endogenous and exogenous expressed proteins/peptides. E.coli BL21 (DE3) have not been optimized for disulfide bond formation. Authors must confirm their assay with peptides expressed in bacteria modified to allow the stable formation of disulphide bonds in their cytoplasm.

Response:   The authors perceive the critics regarding the MBP-ED3 expression in E. coli but we have used a periplasmic not a cytoplasmic expression system.

                   We used MBP as solubility and affinity tag for ED3 using pMAL-p4x expression plasmid for periplasmic expression in E. coli

- MBP is a periplasmic protein

- the reducing environment of the cytoplasm is bypassed by malE in pMal-p4x. This gene encodes a signal sequence that can direct MBP to the bacterial periplasm.

- p  in p4x stands for periplasmic expression. (pMal-c4x would be cytoplasmic).

- BL21(DE3) is a recommended strain in the Klint et al., reference and is also chosen by us because it is protease deficient. For example gor- or trxB- strains are for a reducing cytoplasmic environment and therefore such strains will be good for cytoplasmic expression. The MBP tag is the alternative periplasmic approach.

                   In summary, we had the best results with the method described in the manuscript also recommended by a large body of literature.

                   Please see this reference:

                   Production of recombinant disulfide-rich venom peptides for structural and functional analysis via expression in the periplasm of E. coli. Klint JK1, Senff S, Saez NJ, Seshadri R, Lau HY, Bende NS, Undheim EA, Rash LD, Mobli M, King GF. PLoS One. 2013 May 7;8(5):e63865. doi: 10.1371/journal.pone.0063865. Print 2013.

Reviewer:   - Overexpressed peptides have not been optimized for bacterial expression and there might be some issues regarding rare codons. Authors must verify that their expression system assures the proper production of their peptides of interest.

Response:   Each batch of E. coli culture used for the expression and purification of antigens was controlled by DNA sequencing to make sure that the antigens were not mixed up. We have also tested ED3 antigens using ascites Abs that were DENV-type specific as a control (data not shown).

Codon optimization was tested but was not successful and MBP-ED3 expression was lowered in total or the proportion of MBP-ED3 was much lower than MBP itself. Thus, based on our experience we prefer DENV wt-sequences over codon adapted ones.

                   We also tested Rosetta  (which expresses rare tRNAs). But this led to a much lower yield of MBP-fusion proteins and also to no differences in reactivity to DENV sera when expressed in BL21(DE3).

However, it has been demonstrated that codon optimization or randomization will not always give higher levels of expression. Please see this reference:

Goodman DB1, Church GM, Kosuri S. Causes and effects of N-terminal codon bias in bacterial genes. Science. 2013 Oct 25;342(6157):475-9. doi: 10.1126/science.1241934. Epub 2013 Sep 26.

In MBP-fusion proteins the N-terminus is that of MBP and is identical for all the antigens. Since the codon usage optimization is still complex, we prefer Flavi wt-sequences linked behind MalE. (By the way: The Flavi genom is organized more like genes in prokaryotes, only transcribed and translated in the cytoplasm without the splicing machinery. It would be interesting to figure out, in regard to codon usage and extensive virus variation, why these genes can be efficiently expressed in the nucleus of eukaryotic cells? But this is another question.) Altogether, we have not found any problems using wt sequences in E. coli but we had problems with the codon optimized versions.

Reviewer:   As the study rational only relies on the ED3 dot assay, authors must clarify these two latter points to access publication in Viruses.

                   2) English writing is poor and must be improved. This makes the manuscript track not easy to follow and the understanding not optimal. For example, the following section is absolutely not understandable. What do the authors mean? This is an important section of the manuscript to understand the following sample analysis and it really lacks of pedagogy.

                   « Reactivity of DENV-positive sera against ED3 antigens showed under the experimental conditions (Figure 2, serum dilution 1:100) that in the presented set of (m)ED3 positives, no serotype-specific reactivity was observed. In contrast, the mED3 antigen panel reacted more specific and we observed single, double, triple and quadrupole serotype-specific antibody responses. »

Response:   We thank the reviewer for noticing this mED3>ED3 typing error. And yes, quadruple mED3 reactivity is not serotype-specificity. The text is now:

                   Reactivity of DENV-positive sera against ED3 and ED3s antigens showed under the experimental conditions no serotype-specific reactivity (Figure 4, serum dilution 1:100). Most of the ED3s or ED3 Dengue-specific dots reacted to sera and especially all the ED3 antigens were highly reactive. ED3 for JEV and WNV also showed strong responses whereas sera were negative to the TBEV antigen. One of the sera showed a weak response to the MBP control but clearly distinguished between the four mED3 antigens (double mED3 positive for DENV-3, -4, negative for DENV-1, -2). In contrast to ED3 and ED3s, the mED3 antigen panel reacted more specific and we observed single, double, triple positive dot results with also clear negative dot results. Also quadruple responses were detected by mED3 antigens (Figue 4, all). These sera reacted also strongly to the other 8 DENV-1-4 ED3 and ED3s antigens. Thus, in these overall reactive sera no serotype-specific response could be detected by mED3, In Table 3 direct and indirect diagnostic parameters of the sera used in Figure 4 are shown.

Reviewer:   Minor points:

                   1) « Since the majority of the antibodies are not serotype-specific [26], ED3 antigens are rarely used for the detection of serotype-specific anti-DENV-antibodies [27,28]. Altogether, the specificity of ED3-derived antigens for the determination of DENV serotypes and their application in a serological test seems to be controversial and needs further clarification. »

Authors must temperate their assumption that ED3 epitopes have not been related to DENV-specific antibodies as a large body of literature is showing that ED3 induces sero-specific rather than cross-reactive antibodies.

Response:   To our understanding, the study of human mAB to DENV E showed that they were to 99% fully or partially cross reactive. Dejnirattisai et al. (26) have also shown that only 1% of anti-E is serotype-specific. This is what we wanted to say by … the majority of the antibodies …..

                   We agree that ED3 induced serotype specific responses. However, we have studied antigenicity of ED3 antigens and not their immunogenicity which makes a huge difference.

Additionally, the human Ab serotype study by Dejnirattisai et al. was carried out by western-blot after SDS-Page and a dot blot assay, which is technically very similar to our dot assay.

The authors agree that the phrasing can lead to a misleading interpretation. The text is now changed to:

   As ED3 incorporates so many epitopes for a broad range of antibodies, this domain is frequently used as antigen in diverse serological assays [22–25]. However, studies on the natural immune response to DENV based on the characterization of human monoclonal antibodies (mAb) against the E protein demonstrated that the majority of the antibodies was fully or partially cross-reactive and only 1% of the anti‑E human mAbs were serotype-specific [26]. Additionally, human  mAb were predominantely directed to E proteins on the virion surface and not to recombinant E  [27]. Thus, recombinant E or ED3 antigens are rarely used for the detection of serotype-specific anti‑DENV-antibodies from natural infection [28,29].

Reviewer:   2) Figure 1B- highlighting differences in the DENV sequences would be helpful for the readers.

Response:   has been changed

Reviewer:   3) Typing errors occur along the manuscript. Make sure that appropriate changes will be made in the revised manuscript.

Response:   has been changed

Reviewer:   4) Please update the authors details (Buchy P).

Response:   has been changed

Reviewer 3 Report

A very well written paper describe a new method to assess humoral immunological response to the infection of the different Dengue serotype. the describe method is easy and rapid to perform if compared to PRNT and can support seroprevalence studies and investigation aimed to unravel the complex interaction of antibody production due to reinfection with different dengue serotypes. Further, the data provided could be useful also for the development of further Ab detection method based on D3 domain. 

I have only a minor suggestion:

Line 53 I do not think that Dengue virus isolation can be still considered a “preferential diagnostic method” for DENV acute infection, virus detection is now based mainly on RT-PCR and NS1 Ag detection

Author Response

Response to the comments of Reviewer 3:

Reviewer:   A very well written paper describes a new method to assess humoral immunological response to the infection of the different Dengue serotype. The described method is easy and rapid to perform if compared to PRNT and can support seroprevalence studies and investigation aimed to unravel the complex interaction of antibody production due to reinfection with different dengue serotypes. Further, the data provided could be useful also for the development of further Ab detection method based on D3 domain. 

I have only a minor suggestion:

Line 53 I do not think that Dengue virus isolation can be still considered a “preferential diagnostic method” for DENV acute infection, virus detection is now based mainly on RT-PCR and NS1 Ag detection

Response:   We thank the reviewer for the useful comment. The sentence is now:

…….The direct virus diagnostic in the clinical specimen is performed by detection of virus antigen (NS1) or viral nucleic acids (RT-PCR), but this is only…….

Round 2

Reviewer 2 Report

Authors should add information about periplasmic processing of MBP.

Author Response

Reviewer:

Authors should add information about periplasmic processing of MBP.

Response:

We have added the requested information to the Materials and Methods section incl. the Klint et al. reference

lanes 143-148:

After BamHI and HindIII digestion, the PCR fragment was cloned into pMAL-p4x vector (New England Biolabs) downstream to the malE gene [31]. Since this gene encodes the malEss signal sequence that directs the maltose binding protein (MBP) to the bacterial periplasm, the reducing environment of the cytoplasm is bypassed to form stable disulfide bonds between the conserved cysteine amino acids present in ED3.